# LLM-Augmented Knowledge Representation Learning via Attention for Knowledge Graph Completion

## Abstract

Knowledge Graph Completion (KGC) is a critical task, yet its performance is often hindered by the data sparsity problem arising from the long-tail distribution of entities. While existing works attempt to enrich representations by incorporating auxiliary information like entity descriptions, this kind of implicit learning approaches often proved ineffective due to the introduction of irrelevant noise. To address this, we propose a novel framework LAKRA, which shifts the paradigm from implicit knowledge encoding to explicit data augmentation. LAKRA leverages a Large Language Model (LLM) to proactively reason and generate high-quality, schema-compliant triples for sparse entities, mitigating data sparsity at its source. Besides, we design a powerful encoder-decoder architecture for representation learning, which features a query-aware hybrid attention encoder and a deep feature interaction decoder to capture complex structural and semantic patterns. Experiments conducted on the benchmark datasets demonstrate that LAKRA achieves highly competitive performance on link prediction tasks involving infrequent entities. Our work presents an effective new paradigm for tackling data sparsity in knowledge graphs.

## 1 Introduction

Knowledge Graphs (KGs), such as Freebase and WordNet, have become a cornerstone of modern artificial intelligence, organizing structured factual knowledge in the form of triples $(h, r, t)$. These graphs serve as the backbone for a myriad of downstream applications, including question answering (Hao et al., 2022), recommendation systems (Qin et al., 2023), and semantic search. Despite their large scale, KGs are notoriously incomplete, thus, the task of Knowledge Graph Completion (KGC) has been proposed inferring these missing links. We divide existing KGC methods into three categories, including embedding-based models, NN-based models and LLM-based models.

Early embedding-based models like TransE (Bordes et al., 2013) and RotatE (Sun et al., 2019) learn vector representations for this task, but their performance is heavily dependent on link density, making them vulnerable to data sparsity. To mitigate this, some methods (Wang et al., 2024a; Xue et al., 2021) enrich representations by incorporating auxiliary information like entity descriptions. However, this approach relies on an implicit learning paradigm, tasking the model with distilling useful cues from often noisy textual data, which can increase complexity without guaranteed gains. Neural Network (NN)-based methods such as GNN could better leverage graph topology by aggregating neighborhood information. However, they also struggle with the long-tail distribution of entities, where low-degree "tail" entities lack sufficient neighbors to learn high-quality representations, thus perpetuating data sparsity. Moreover, existing GNN approaches often treat all queries uniformly, failing to adapt their representations to the specific relation being predicted. The recent advent of LLMs has introduced new approaches which we refer to such models as LLM-based models. These models have been explored in various ways, such as serving as encoders to provide richer initial entity representations or being directly prompted to predict missing links in a few-shot or zero-shot manner. Many existing LLM-based approaches treat the model's knowledge as an implicit source to be encoded or fine-tuned, failing to solve the graph's underlying structural sparsity. Moreover, due to the massive parameter scale of the models, these approaches also incur high computational costs.

To bridge these gaps, we propose LAKRA, a framework that shifts the paradigm from implicit encoding to explicit data augmentation. We firstly employ an LLM stage to generate high-quality,

schema-compliant triples for sparse entities, directly tackling the long-tail problem. This enriched graph is then processed by LAKRA's powerful encoder-decoder architecture, which features a query-aware hybrid attention encoder and a 3D convolutional decoder. Our main contributions are as follows:

- We propose a novel framework that leverages Large Language Models for explicit data augmentation to address the data sparsity problem in knowledge graphs, generating high-quality, schema-aware triples for infrequent entities.
- We design a powerful encoder-decoder model, LAKRA, which features a query-aware hybrid attention encoder for learning expressive representations to address the issue of indiscriminate neighborhood aggregation by focusing only on relevant information for a given query and a 3D convolutional decoder for capturing deep, multi-level feature interactions.
- We conduct extensive experiments on benchmark datasets, demonstrating that our proposed framework achieves highly competitive performance, particularly in predicting links involving sparse entities.

## 2 RELATED WORK

Embedding-based models learn low-dimensional vector representations for entities and relations. Translational models interpret relations as geometric operations, such as translation in TransE, projection in TransH and TransR (Lin et al., 2015), or rotation in the complex space in RotatE. Tensor factorization models frame KGC as a 3D tensor completion problem. RESCAL uses a full matrix for each relation, which DistMult simplifies to a diagonal matrix, and ComplEx (Trouillon et al., 2016) extends to the complex space to handle asymmetry. To better model hierarchical structures, non-Euclidean models like MuRP (Balazevic et al., 2019) and AttH (Liu et al., 2019) leverage hyperbolic geometry. While computationally efficient, embedding-based models are fundamentally limited by data sparsity, as their reliance on observed triples alone hinders their ability to model complex graph structures.

Another line of work enriches representations by incorporating auxiliary information. Models like TAPR (Wang et al., 2014), SSP (Xiao et al., 2017), TKRL (Xie et al., 2016), and MTE (Wang et al., 2021) integrate textual descriptions and type hierarchies to enhance embeddings. Similarly, KG-BERT (Yao et al., 2019) reframes link prediction as a sequence classification task for language models, relying on the model to implicitly distill facts from text.

Neural network architectures are used to capture more complex patterns. CNN-based models focus on local interactions. ConvE reshapes embeddings into 2D "images" for convolution, a technique enhanced by InteractE (Vashishth et al., 2020) with feature permutation and circular convolution. ConvKB (Nguyen et al., 2017) applies 1D convolution over concatenated triple embeddings. GNN-based models learn representations by aggregating neighborhood information. R-GCN uses relation-specific matrices, while GAT introduces dynamic attention weights to differentiate neighbor importance. CompGCN (Vashishth et al., 2019) improves parameter efficiency by using composition operators during message passing. SACN (Shang et al., 2019) combines a GNN encoder with a convolutional decoder. While powerful, GNNs are fundamentally limited by neighborhood density, a key issue our data augmentation strategy is designed to resolve.

LLMs are increasingly used in KGC due to their vast parametric knowledge. One approach uses LLMs as feature encoders, such as KG-BERT, which fine-tunes BERT on textualized triples. iHT (Chen et al., 2023) initializes its entity encoder with pre-trained BERT weights to create rich entity representations for knowledge graph completion. Another reframes KGC as a generation or question-answering task, either by prompting large-scale LLMs or by fine-tuning models like KGT5 (Saxena et al., 2022) to inject structural knowledge. These methods use LLMs implicitly at inference time. In stark contrast, our work employs an LLM in an explicit data augmentation role during a pre-processing stage to directly enrich the graph structure itself. RAA-KGC (Yuan et al., 2025) enhances PLM-based knowledge graph completion by augmenting input queries with relation-aware anchor entities to create a more discriminative representation for link prediction.

## 3 METHODOLOGY

In this section, we introduce our proposed model for knowledge graph completion(KGC), named LAKRA. LAKRA is designed to mitigate data sparsity via explicit, LLM-based data augmenta-

tion and then capture both deep relational semantics and complex structural patterns within the knowledge graph. The overall architecture, depicted in Figure 1, consists of three main components: (1)LLM-based data augmentation for sparse entities, (2) a novel query aware graph attention encoder for learning expressive entity and relation representations, and (3) a 3D deep feature interaction decoder for link prediction.

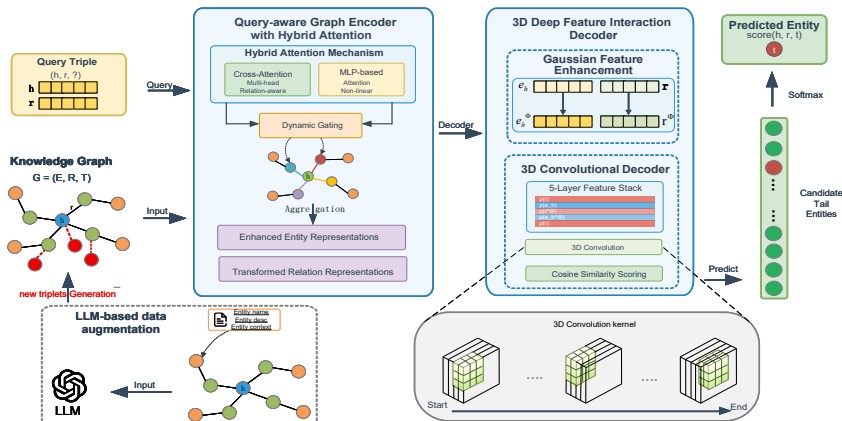

Figure 1: Overall architecture of the LAKRA framework showing the LLM-based data augmentation module and the encoder-decoder architecture.

### 3.1 PROBLEM FORMULATION

A knowledge graph (KG) is denoted as $\mathcal{G} = (\mathcal{E}, \mathcal{R}, \mathcal{T})$, where $\mathcal{E}$ is the set of entities, $\mathcal{R}$ is the set of relations, and $\mathcal{T}$ is the set of factual triples $(h, r, t)$, with $h, t \in \mathcal{E}$ representing the head and tail entities, and $r \in \mathcal{R}$ representing the relation. The task of KGC aims to predict a missing entity, i.e., $t$ given $(h, r, ?)$ or $h$ given $(?, r, t)$.

### 3.2 LLM-BASED DATA AUGMENTATION FOR SPARSE ENTITIES

We propose a novel LLM-based data augmentation methodology to mitigate data sparsity arising from the long-tail distribution in knowledge graphs. Our core idea is to guide an LLM with the existing information and graph-native structural priors to generate high-quality, schema-compliant new triples for "tail entities." The methodology comprises three main stages: tail entity identification and augmentation targeting, type-Constrained candidate set construction, two-stage generation via LLMs. Technical details are as follows:

**Tail Entity Identification and Augmentation Targeting.** Knowledge graphs exhibit a long-tail distribution where most entities appear infrequently, hindering model performance.To precisely target the entities most in need of augmentation, we first calculate the dgree $f(e)$ of each entity $e$ in the training set $\mathcal{T}_{train}$. The degree $f(e)$ is defined as the degree of the entity $e$ as either a head or a tail in $\mathcal{T}_{train}$.

$$f(e) = \mid \{(h, r, t) \in \mathcal{T}_{\text{train}} \mid h = e \vee t = e\} \mid \tag{1}$$

We rank all entities in descending order based on their degree $f(e)$ and select the bottom 20% as our target set for augmentation. This set, denoted as $\mathcal{E}_{train}$, represents the entities most adversely affected by data sparsity.

**Type-Constrained Candidate Set Construction.** To ensure the generated triples are both semantically plausible and structurally consistent with the knowledge graph's in-trinsic schematic properties, we construct two type-constrained candidate sets: a Relation Candidate Table $\mathcal{C}_{rel}$ and an Entity Candidate Table $\mathcal{C}_{ent}$. $\mathcal{C}_{rel}$ maps each entity type to its frequently co-occurring relations (as head or tail), while $\mathcal{C}_{ent}$ maps each relation to its common head and tail entity types, subject to a minimum frequency threshold.

$$\mathcal{C}_{rel}(t_e, \text{head}) = \{r \mid \exists (h, r, t) \in \mathcal{T}_{train}, \text{type}(h) = t_e, f_{r_h} > min\} \tag{2}$$

$$\mathcal{C}_{rel}(t_e, \text{tail}) = \{r \mid \exists (h, r, t) \in \mathcal{T}_{train}, \text{type}(t) = t_e, f_{r_t} > min\} \tag{3}$$

where $f_{r_h}$ and $f_{r_t}$ represent the frequencies of the corresponding relations when the entity type $t_e$ serves as either the head entity or the tail entity.

$$\mathcal{C}_{ent}(r', \text{head}) = \{t_h \mid \exists (h, r, t) \in \mathcal{T}_{train}, r = r', f_h > min\} \tag{4}$$

$$\mathcal{C}_{ent}(r', \text{tail}) = \{t_t \mid \exists (h, r, t) \in \mathcal{T}_{train}, r = r', f_t > min\} \tag{5}$$

where $f_h$ and $f_t$ denote the frequencies of the types of corresponding head or tail entities when the relation is $r'$.

These candidate tables (example illustrated conceptually in Figure 2) serve as a form of "soft constraint," furnishing the subsequent LLM generation phase with structured, graph-native prior knowledge.

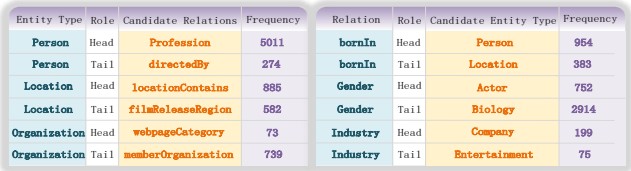

| Entity Type | Role | Candidate Relations | Frequency | | Relation | Role | Candidate Entity Type | Frequency |
|---|---|---|---|---|---|---|---|---|
| Person | Head | Profession | 5011 | | bornIn | Head | Person | 954 |
| Person | Tail | directedBy | 274 | | bornIn | Tail | Location | 383 |
| Location | Head | locationContains | 885 | | Gender | Head | Actor | 752 |
| Location | Tail | filmReleaseRegion | 582 | | Gender | Tail | Biology | 2914 |
| Organization | Head | webpageCategory | 73 | | Industry | Head | Company | 199 |
| Organization | Tail | memberOrganization | 739 | | Industry | Tail | Entertainment | 75 |

Figure 2: An example of type-constrained candidate set. The relation candidate table $\mathcal{C}_{rel}$ stores relations associated with each entity type, while the entity candidate table $\mathcal{C}_{ent}$ stores entity types associated with each relation.

**Two-Stage Generation via LLMs.** For each tail entity $e_{tail} \in \mathcal{E}_{tail}$, we employ a two-stage generation process to create new triples, including relation generation and entity generation. In relation

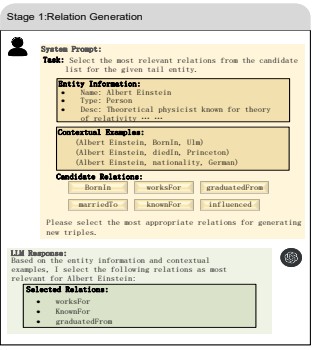 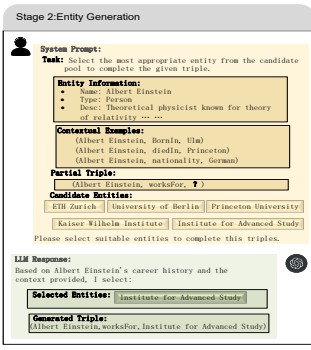

Figure 3: Two-stage LLM-based triple generation process. (a) Stage 1 generates candidate relations based on entity type constraints. (b) Stage 2 generates target entities based on relation and type constraints.

generation stage, we aim to identify plausible relations given entity $e_{tail}$. We handle cases where $e_{tail}$ acts as a head and as a tail entity separately. Specifically, given a tail entity $e_{tail}$, we first retrieve a list of candidate relations from the Relation Candidate Table $\mathcal{C}_{rel}$ based on its type, type($e_{tail}$). Subsequently, we provide the LLM with a structured prompt, $\mathcal{P}_{rel}$ (shown in Figure 3), containing a task description, entity information (name, type, description), contextual examples, and the candidate relation list from $\mathcal{C}_{rel}(\text{type}(e_{tail}), \text{head})$. The LLM output a subset of relations, $\mathcal{R}_{pred}$, which are most likely to form a factual statement with $e_{tail}$.

$$\mathcal{R}_{pred} = \text{LLM}\left(\mathcal{P}_{rel}\left(e_{tail}, \mathcal{C}_{rel}\right)\right) \tag{6}$$

Entity generation stage, for each predicted relation $r_{pred} \in \mathcal{R}_{pred}$, finds suitable entities to complete the triple $(e_{tail}, r_{pred}, ?)$ or $(?, r_{pred}, e_{tail})$. We first determine the common entity types for the relation $r_{pred}$ from an Entity Candidate Table $\mathcal{C}_{ent}$ and create a candidate entity pool of those types. A new prompt similar to $\mathcal{P}_{rel}$, $\mathcal{P}_{ent}$ (shown in Figure 3), is designed, containing the partial triple and a candidate entity list constrained by $\mathcal{C}_{ent}(r_{pred}, \text{tail})$. The LLM then selects the most suitable tail entities $t_{pred}$ from candidate entity list, thereby forming a complete generated triple $(e_{tail}, r_{pred}, t_{pred})$.

$$t_{pred} = \text{LLM}\left(\mathcal{P}_{ent}\left(e_{tail}, r_{pred}, \mathcal{C}_{ent}\right)\right) \tag{7}$$

Through this two-stage process, we generate a set of new candidate triples, $\mathcal{T}_{gen}$, for each tail entity and we filter out any triples that already exist in the original training set to ensure the novelty of the augmented knowledge. Conventional knowledge graph enhancement techniques often seek to address data sparsity by enriching entity representations. These methods typically integrate auxiliary information, such as entity names, descriptions, and types, into the learning process, compelling the completion model to implicitly decipher relational patterns from this encoded knowledge. The efficacy of such an approach is indirect, as it remains uncertain whether the model can effectively capitalize on this supplementary context. In a distinct departure, our methodology re-frames this challenge from one of implicit representation learning to one of explicit knowledge generation. We leverage potent reasoning capabilities of an LLM to proactively distill salient knowledge and materialize it as structured triples, offering a more direct and verifiable approach to data augmentation.

### 3.3 QUERY-AWARE GRAPH ATTENTION ENCODER WITH HYBRID ATTENTION

Our encoder learns expressive, query-aware entity representations by aggregating neighborhood information guided by the specific query relation, departing from traditional GNNs that handle message passing uniformly. To capture a wide range of interaction patterns, we introduce a hybrid attention mechanism that integrates two parallel components: (1) Query-aware Multi-head Cross Attention and (2) MLP-based Additive Attention. Given a triple $(e_i, r_{ij}, e_j)$, the attention score $\alpha_{ij}$ between the central node $i$ and a neighbor node $j$ is computed as follows.

**Query-aware Multi-head Cross Attention.** This mechanism is designed to filter irrelevant context by making the aggregation specific to the link prediction query. Specifically, for each attention head, the query $\mathbf{Q}$ is derived from the central entity and the query relation, while the key $\mathbf{K}$ is derived from neighboring entities and their connecting relations:

$$\mathbf{Q}^{(k,l)} = \mathbf{W}_q^{(k,l)}(\mathbf{W}_{combine}[\mathbf{e}_i^{(l)} \oplus \mathbf{r}_q^{(l)}]) \tag{8}$$

$$\mathbf{K}^{(k,l)} = \mathbf{W}_k^{(k,l)}\mathbf{e}_j^{(l)} + \mathbf{W}_{rel}^{(k,l)}\mathbf{r}_{ij}^{(l)} \tag{9}$$

where $\mathbf{W}_q^{(k,l)}, \mathbf{W}_k^{(k,l)}, \mathbf{W}_{rel}^{(k,l)} \in \mathbb{R}^{d' \times d}$ are the dedicated projection matrices for the $k$-th head in the $l$-th layer. $\mathbf{e}_i^{(l)}$ is the central entity and $\mathbf{e}_j^{(l)}$ is a neighboring entity at the $l$-th layer.

The resulting scaled cross attention score is calculated with a learnable, relation-specific bias:

$$s_{dot}^{(k,l)} = \frac{(\mathbf{Q}^{(k,l)})^T \mathbf{K}^{(k,l)}}{\sqrt{d'}} + \mathbf{b}_{r_{ij}} \tag{10}$$

This formulation captures complex intrinsic connections among entities, relations and queries to discover "semantic relevance".

**MLP-based Additive Attention.** The alternative attention mechanism involves concatenating the embeddings of the central entity $\mathbf{e}_i$, the neighboring entity $\mathbf{e}_j$, and the relation between them $\mathbf{r}_{ij}$, which is then passed through an MLP layer. This non-linear transformation enables the learning of more complex, localized feature combinations and even structurally complementary correlations. The computation is as follows:

$$s_{mlp}^{(l)} = \mathbf{w}_a^{(l)} \text{LeakyReLU}(\mathbf{W}_{att}^{(l)}[\mathbf{e}_i^{(l)} \oplus \mathbf{r}_{ij}^{(l)} \oplus \mathbf{e}_j^{(l)}] \tag{11}$$

where $\oplus$ denotes the concatenation operation, and $\mathbf{W}_{att} \in \mathbb{R}^{d \times 3d}$ and $\mathbf{w}_a \in \mathbb{R}^{1 \times d}$ are learnable weight matrices at the $l$-th layer.

**Hybrid Attention.** To capture a comprehensive set of relational patterns, our final attention score is derived from a dynamic fusion of the cross-attention and MLP-based attention paradigms. This is achieved through a gating mechanism that adaptively weighs the two components for each edge based on the context of its nodes. This design endows the model with the flexibility to handle diverse neighborhood dependencies more effectively than any single-mechanism approach. The dynamic gating mechanism is calculated as follows:

$$g_{ij}^{(l)} = \sigma(\mathbf{W}_g^{(l)}[\mathbf{e}_i^{(l)} \oplus \mathbf{e}_j^{(l)}]) \tag{12}$$

$$s_{irj}^{(l)} = g_{ij}^{(l)} \left( \frac{1}{K} \sum_{k=1}^{K} s_{dot}^{(k,l)} \right) + (1 - g_{ij}^{(l)}) s_{mlp}^{(l)} \tag{13}$$

Here, $s_{irj}^{(l)}$ is the final attention score. The gate weight $g_{ij}$ is dynamically computed from the embeddings of the central node $\mathbf{e}_i^{(l)}$ and the neighboring node $\mathbf{e}_j^{(l)}$, representing the weights assigned to the two attention mechanisms, which sum to one. Then, each attention score is normalized using softmax:

$$\alpha_{irj}^{(l)} = \text{softmax}_{rj}\left(s_{irj}^{(l)}\right) = \frac{\exp\left(s_{irj}^{(l)}\right)}{\sum_{j' \in \mathcal{N}_i} \sum_{r' \in \mathcal{R}_{ij}} \exp\left(s_{ir'j'}^{(l)}\right)} \tag{14}$$

where $\mathcal{N}_i$ is the set of neighboring entities of entity $i$, $\mathcal{R}_{ij}$ is the set of relations connecting entity $i$ and entity $j$, and $\alpha_{irj}^{(l)}$ represents the final normalized attention weight at the $l$-th layer.

**Aggregation.** Inspired by advanced models (Nickel et al., 2016), we employ the circular cross-correlation function, denoted as $corr(\mathbf{e}_j, \mathbf{r}_{ij})$, to fuse entity and relation embeddings. Unlike simpler element-wise operators, this method models comprehensive, dimension-level interactions between vectors using Fast Fourier Transform. Then the message function is defined below:

$$m(\mathbf{e}_j, \mathbf{r}_{ij}) = \boldsymbol{W}_v corr(\mathbf{e}_j, \mathbf{r}_{ij}) \tag{15}$$

where $\boldsymbol{W}_v$ is a projection matrix that corresponds to the previously mentioned $\boldsymbol{W}_q$ and $\boldsymbol{W}_k$.

Thus, with $m(\mathbf{e}_j, \mathbf{r}_{ij})$ acting as the aggregated message and value V corresponding to query Q and key K, the final entity representation is updated as follows:

$$\mathbf{e}_i^{(l+1)} = Tanh\left(\sum_{(e_j^{(l)}, r_{ij}^{(l)}) \in \mathcal{N}_i} \alpha_{irj}^{(l)} m(\mathbf{e}_j^{(l)}, \mathbf{r}_{ij}^{(l)}) + \boldsymbol{W}_o^{(l)} \mathbf{e}_i^{(l)}\right) \tag{16}$$

where $\alpha_{irj}^{(l)}$ is the attention weight for the contextual triple. To prevent issues of gradient vanishing and over-smoothing, we have also incorporated a residual connection.

After the final entity embedding is obtained, the relation representation undergoes a corresponding transformation, as follows:

$$\mathbf{r}^{(l+1)} = \mathbf{W}_{update}^{(l)} \cdot \mathbf{r}^{(l)} \tag{17}$$

where $\mathbf{W}_{update}^{(l)} \in \mathbb{R}^{d \times d}$ is a learnable weight matrix for relation at the $l$-th layer.

### 3.4 3D DEEP FEATURE INTERACTION DECODER

Our deep feature interaction decoder is designed to capture complex relational patterns. Gaussian feature enhancement module generates a non-linear, augmented feature space from the original embeddings and 3D convolutional network processes a multi-view stack of both the original and enhanced representations to model their deep, multi-level interactions for link prediction.

**Gaussian Feature Enhancement.** This module projects the original entity and relation embeddings into a higher-dimensional space using a set of learnable Gaussian kernels $\phi_k(\mathbf{x})$. The intuition is to capture the uncertainty and create a non-linear feature space that highlights specific characteristics of the embeddings based on their proximity to learned "concept centers".

For an embedding $\mathbf{x}$ (either an entity $\mathbf{e}_h$ or relation $\mathbf{r}$), its Gaussian-mapped representation $\mathbf{x}^\Phi$ is computed as:

$$\mathbf{x}^\Phi = [\phi_1(\mathbf{x}), \phi_2(\mathbf{x}), \ldots, \phi_N(\mathbf{x})] \tag{18}$$

where $\phi_k(\mathbf{x})$ is the $k$-th kernel function.

**3D Convolutional Decoder.** For link prediction, we utilize a 3D convolutional network inspired by the work of (Zhang et al., 2024) to model deep feature interactions. First, a 5-layer 3D feature stack $\mathbf{S} \in \mathbb{R}^{5 \times H \times W}$ is constructed from the original and Gaussian-enhanced embeddings of the head entiy $\mathbf{e}_h$ and relation $\mathbf{r}$:

$$\mathbf{S} = \text{stack}([\rho(\mathbf{r}), \rho(\mathbf{e}_h), \rho(\mathbf{r}^\Phi), \rho(\mathbf{e}_h^\Phi), \rho(\mathbf{r})], \text{axis=2}) \tag{19}$$

This multi-view stack is then processed by a 3D convolution using a kernel $\mathbf{\Omega} \in \mathbb{R}^{C \times 5 \times k_w \times k_h}$ to produce an interaction vector $\mathbf{v}_{hr}$:

$$\mathbf{v}_{hr} = f(\text{vec}(\mathbf{S} * \mathbf{\Omega})\mathbf{W}_p) \tag{20}$$

where $*$ denotes the 3D convolution and $\text{vec}()$ is the vectorization operation.

A key design element is the convolutional kernel, which simultaneously processes two adjacent layers. As the kernel traverses the depth and spatial dimensions ($k_h \times k_w$), it systematically evaluates four interaction pairings in a single pass, original-original ($\mathbf{e}_h, \mathbf{r}$), original-enhanced ($\mathbf{e}_h, \mathbf{r}^\phi$), enhanced-enhanced ($\mathbf{e}_h^\phi, \mathbf{r}^\phi$), and enhanced-original ($\mathbf{e}_h^\phi, \mathbf{r}$). This approach captures rich, multi-level interactions across different representational spaces, offering a more comprehensive modeling capacity than standard 2D methods.

The final score for a candidate triple $(h, r, t)$, $\text{score}(h, r, t)$, is computed as the cosine similarity between the resulting interaction vector $\mathbf{v}_{hr}$ and the candidate tail entity embedding $\mathbf{e}_t$.

## 3.5 TRAINING OBJECTIVE

For the primary link prediction task, we use the standard cross-entropy loss, which aims to maximize the score of the true tail entity over all other entities:

$$\mathcal{L} = -\frac{1}{B} \sum_{i=1}^{B} \left( \log(\sigma(\text{score}(h_i, r_i, t_i))) + \sum_{j=1}^{N_s} \log(1 - \sigma(\text{score}(h_i, r_i, t'_{ij}))) \right) \tag{21}$$

where $B$ denotes the batch size, $(h_i, r_i, t_i)$ represents the $i$-th positive triple in the batch, $(h_i, r_i, t'_{ij})$ denotes the $j$-th negative triple corresponding to the $i$-th positive sample. $N_s$ represents the number of negative samples generated for each positive triple. The total loss is a weighted sum of the two components:

$$\mathcal{L}_{total} = \mathcal{L}_{orig} + \lambda \mathcal{L}_{aug} \tag{22}$$

where $\lambda$ is a hyperparameter balancing the two objectives. $\mathcal{L}_{orig}$ denotes the loss term computed from original datasets. $\mathcal{L}_{aug}$ denotes the loss term computed solely from triples generated by LLMs.

## 4 EXPERIMENT

**Dataset.** To evaluate the performance of our model comprehensively and ensure fair comparison with existing methods, we conduct experiments on two widely adopted, challenging benchmark datasets for knowledge graph completion. Both datasets exhibit complex relational patterns essential for rigorously assessing KGC models. Dataset statistics are showed in Table 1.

**Evaluation Metrics.** Our evaluation framework uses four complementary ranking-based metrics to assess different aspects of model performance. The main metric is Mean Reciprocal Rank (MRR), which is the average of the reciprocal ranks of the correct answers. We also report Hits@K for K values of 1, 3, and 10, showing the percentage of correct entities ranked within the top-K predictions. Our evaluation uses a "filtered" setting, which removes all other known true triples from the ranking process to account for the incompleteness of knowledge graphs.

**Baseline.** To provide a thorough evaluation, we compare our proposed model against a comprehensive set of competitive baselines, categorized into three main paradigms: Embedding-based models including TransE, RotatE, HAKE(Zhang et al., 2020), QuatRE(Nguyen et al., 2022) and CompilE(Cui & Zhang, 2024), GNN- based models incluNetwork ding R-GCN(Schlichtkrull et al., 2018), ConvE, InteractE(Vashishth et al., 2020), BiGAT(Xie et al., 2021) and RHKH(Wang et al., 2024b), and LLM-based models including KG-BERT(Yao et al., 2019), KG-R3(Pahuja et al., 2023), iHT(Chen et al., 2023) and RAA-KGC(Yuan et al., 2025).

Table 1: Base information about experimental Datasets.

| Dataset | #ent | #rel | #train | #valid | #test |
|---|---|---|---|---|---|
| FB15k-237 | 14,541 | 237 | 272,115 | 17,535 | 20,466 |
| WN18RR | 40,493 | 11 | 86,835 | 3,034 | 3,134 |

**Link Prediction Task.** Table 2 presents the link prediction performance of our model against state-of-the-art baselines. The results show that LAKRA is a highly competitive framework, consistently achieving top-tier performance.

Table 2: Main experimental results on FB15K-237 and WN18RR datasets. The best results are highlighted in bold and the second best results are underlined, '-' indicates the result is not reported in previous work.

| Model | | FB15K-237 | | | | WN18RR | | | |
|---|---|---|---|---|---|---|---|---|---|
| | | MRR | Hits@1 | Hits@3 | Hits@10 | MRR | Hits@1 | Hits@3 | Hits@10 |
| Embedding Based Model | TransE | 0.287 | 0.192 | 0.325 | 0.475 | 0.243 | 0.043 | 0.441 | 0.532 |
| | RotatE | 0.338 | 0.241 | 0.375 | 0.533 | 0.494 | 0.455 | 0.510 | 0.571 |
| | HAKE | 0.346 | 0.250 | 0.381 | 0.542 | 0.497 | 0.452 | 0.516 | 0.582 |
| | QuatRE | 0.367 | 0.269 | 0.404 | 0.563 | 0.493 | 0.439 | 0.519 | 0.592 |
| | CompilE | 0.372 | 0.277 | 0.408 | 0.563 | 0.495 | 0.453 | 0.510 | 0.579 |
| Neural Network Based Model | R-GCN | 0.248 | 0.158 | 0.275 | 0.428 | 0.123 | 0.080 | 0.137 | 0.207 |
| | ConvE | 0.316 | 0.239 | 0.350 | 0.491 | 0.460 | 0.390 | 0.430 | 0.480 |
| | InteractE | 0.354 | 0.263 | - | 0.535 | 0.463 | 0.430 | - | 0.528 |
| | BiGAT | 0.366 | 0.270 | 0.402 | 0.558 | 0.505 | 0.458 | 0.521 | 0.594 |
| | RHKH | 0.353 | 0.271 | - | - | 0.480 | 0.445 | - | - |
| PLM Based Model | KG-BERT | 0.268 | 0.197 | 0.289 | 0.420 | 0.438 | 0.412 | 0.465 | 0.524 |
| | KG-R3 | 0.390 | 0.310 | 0.413 | 0.539 | 0.467 | 0.430 | 0.479 | 0.536 |
| | iHT | 0.344 | - | - | 0.528 | 0.448 | - | - | 0.678 |
| | RAA-KGC | 0.315 | 0.224 | 0.341 | 0.493 | 0.597 | 0.506 | 0.649 | 0.760 |
| LLM Based Model | KICGPT | 0.412 | 0.327 | 0.448 | 0.554 | 0.549 | 0.474 | 0.585 | 0.641 |
| Our Model | LAKRA | 0.396 | 0.307 | 0.431 | 0.577 | 0.539 | 0.507 | 0.547 | 0.606 |

On this complex, large-scale dataset FB15K-237, LAKRA achieves the best overall performance with top scores on MRR, Hits@3, and Hits@10. Although KG-R3 secures the best Hits@1, our explicit data augmentation provides a more globally complete graph, leading to better overall ranking. Similarly, on the hierarchical WN18RR, while RAA-KGC's anchor-based method performs strongly, LAKRA again achieves the top Hits@1 score, highlighting our model's precision in pinpointing the single correct entity through LLM augmentation and a sophisticated decoder. The strong results across two distinct datasets validate our two-pronged strategy of enriching the data foundation via LLMs and then deeply modeling it with a powerful neural network.

Table 3: Comparison of experimental results under different message functions and decoders. The best results are highlighted in bold and the second best results are underlined.

| Message+Decoder | FB15K-237 | | | | WN18RR | | | |
|---|---|---|---|---|---|---|---|---|
| | MRR↑ | Hits@1 | Hits@3 | Hits@10 | MRR↑ | Hits@1 | Hits@3 | Hits@10 |
| sub+Conv3D | 0.392 | 0.303 | 0.426 | 0.579 | 0.536 | 0.502 | 0.545 | 0.602 |
| corr+Conv3D | 0.396 | 0.307 | 0.431 | 0.577 | 0.539 | 0.507 | 0.547 | 0.606 |
| mult+Conv3D | 0.393 | 0.305 | 0.427 | 0.574 | 0.538 | 0.503 | 0.543 | 0.607 |
| sub+ConvE | 0.383 | 0.293 | 0.417 | 0.560 | 0.524 | 0.488 | 0.536 | 0.596 |
| Corr+ConvE | 0.386 | 0.296 | 0.420 | 0.566 | 0.527 | 0.494 | 0.540 | 0.597 |
| mult+ConvE | 0.384 | 0.295 | 0.418 | 0.567 | 0.525 | 0.488 | 0.538 | 0.594 |
| sub+Transformer | 0.378 | 0.287 | 0.411 | 0.558 | 0.518 | 0.485 | 0.529 | 0.588 |
| corr+Transformer | 0.379 | 0.289 | 0.414 | 0.561 | 0.521 | 0.487 | 0.532 | 0.588 |
| mult+Transformer | 0.379 | 0.288 | 0.413 | 0.560 | 0.517 | 0.486 | 0.530 | 0.586 |

**Performance of Different Decoders and Message Functions.** To validate our architectural choices, we studied the performance of different message functions (sub, mult, corr) and decoders (Conv3D, ConvE, Transformer), with results presented in Table 3.

The results confirm the superiority of our proposed Conv3D decoder, which consistently outperforms the ConvE and Transformer alternatives. Its ability to model complex, multi-level interactions via a 3D feature stack allows it to capture richer relational patterns than 2D convolutions or standard self-attention. Furthermore, circular cross-correlation (corr) proved to be the most effective message function. It achieved the highest MRR and Hits@1 scores, validating its power as a composition operator. Unlike simpler element-wise operations, corr models comprehensive, dimension-level interactions between embeddings, enabling our encoder to capture crucial relational patterns like symmetry and composition.

**Comparison of Different Attention Mechanism Applied in Encoder.** To validate our Hybrid Attention mechanism, we compare it against its standalone components—Cross-Attention and MLP-based Attention, on four relation patterns of FB15K-237, with results presented in Table 4.

This fine-grained analysis demonstrates the superiority of our Hybrid Attention, which achieves the best performance on the more complex 1-1, 1-N, and N-N relations. Notably, the standalone Cross-Attention model excels on N-1 relations, highlighting its strength in capturing clear semantic

regularities. This confirms that no single attention mechanism is universally optimal. Our hybrid model succeeds by adaptively fusing the semantic relevance captured by Cross-Attention with the complex structural patterns from MLP-based Attention, ensuring robust and superior performance across diverse relation types.

**Ablation Studies.** To validate the contribution of each key component in LAKRA, we conducted a series of ablation studies, with results detailed in Table 5.

The study confirms the efficacy of our architectural choices. Replacing our 3D convolutional decoder with a standard ConvE decoder results in the second-largest performance drop, highlighting its superior ability to model deep feature interactions. Furthermore, removing either the MLP-based attention or the cross-attention component leads to a noticeable decline, validating that their dynamic fusion is more effective than either one in isolation.

The most significant finding is the critical importance of our LLM-based data augmentation. Removing the generated triples causes the most substantial performance degradation across all metrics. This severe decline unequivocally demonstrates that our explicit, schema-aware data augmentation strategy is the cornerstone of LAKRA's success. To further investigate this, we analyze the generated triples for two datasets. Interestingly, we found that 932 and 198 of these generated triples for FB15K-237 (10833) and WN18RR (12526), respectively, already existed in their corresponding test sets. Then we conducted an experiment on FB15K-237 where we explicitly removed the 932 overlapping triples from the augmented set and re-trained our model. The results remained highly compelling, proving that the primary benefit comes from enriching the local neighborhoods of sparse entities with novel, plausible triples, not from memorizing test data.

## 5 CONCLUSION

In this paper, we addressed the persistent challenge of data sparsity in knowledge graph completion, a problem rooted in the long-tail distribution of entities. To overcome this, we introduced LAKRA, a novel framework that shifts the paradigm from implicit learning to explicit data augmentation. First, we proposed a schema-aware, LLM-based data augmentation strategy that directly targets and enriches the neighborhood of sparse, tail-end entities by generating high-fidelity triples. Second, we designed a powerful encoder-decoder architecture to effectively learn from this enriched graph. The encoder features a query-aware hybrid attention mechanism to capture a wide spectrum of relational patterns, while the 3D convolutional decoder excels at modeling deep, multi-level feature interactions for accurate link prediction.

Table 4: Comparison of different attention mechanisms on various relation patterns of FB15K-237 dataset. The best results for each pattern are highlighted in bold.

|  | **Hybrid Attention** | | | | **Cross-Attention** | | | | **MLP-based Attention** | | | |
|---|---|---|---|---|---|---|---|---|---|---|---|---|
|  | MRR | H@1 | H@3 | H@10 | MRR | H@1 | H@3 | H@10 | MRR | H@1 | H@3 | H@10 |
| 1-1 | **0.649** | **0.603** | **0.661** | **0.732** | 0.636 | 0.581 | 0.658 | 0.732 | 0.640 | 0.585 | 0.660 | 0.726 |
| 1-N | **0.352** | **0.287** | **0.370** | **0.477** | 0.348 | 0.283 | 0.368 | 0.474 | 0.344 | 0.278 | 0.367 | 0.468 |
| N-1 | 0.493 | 0.438 | 0.512 | 0.595 | **0.498** | **0.441** | **0.523** | **0.601** | 0.494 | 0.440 | 0.514 | 0.593 |
| N-N | **0.362** | **0.257** | **0.405** | **0.574** | 0.354 | 0.248 | 0.395 | 0.568 | 0.355 | 0.248 | 0.399 | 0.567 |

Table 5: Ablation study of LAKRA on the FB15K-237 and WN18RR datasets. 'w/o' denotes the model with the corresponding component removed or replaced. llm-aug denotes the LLM-based data augmentation component.

| Model | **FB15K-237** | | | | **WN18RR** | | | |
|---|---|---|---|---|---|---|---|---|
|  | MRR | H@1 | H@3 | H@10 | MRR | H@1 | H@3 | H@10 |
| w/o $\text{llm}_{aug}$ | 0.364 | 0.272 | 0.396 | 0.547 | 0.505 | 0.469 | 0.518 | 0.577 |
| w/o conv3d | 0.385 | 0.295 | 0.419 | 0.566 | 0.529 | 0.492 | 0.541 | 0.602 |
| w/o mlp | 0.393 | 0.302 | 0.428 | 0.576 | 0.537 | 0.505 | 0.545 | 0.605 |
| w/o cross | 0.391 | 0.301 | 0.426 | 0.574 | 0.535 | 0.503 | 0.547 | 0.604 |
| w/o gauss | 0.389 | 0.304 | 0.427 | 0.571 | 0.532 | 0.503 | 0.544 | 0.603 |
| w/o $\text{llm}_{test}$ | 0.375 | 0.281 | 0.407 | 0.559 | - | - | - | - |
| **LAKRA** | **0.396** | **0.307** | **0.431** | **0.577** | **0.539** | **0.507** | **0.547** | **0.606** |

## 6 ACKNOWLEDGMENTS

We thank ChatGPT-4o for its assistance in proofreading and editing the language of this paper.

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

## A    DETAILS ABOUT DATASETS

Figure 4 and Table 6 illustrate the long-tail distribution of entity degrees in the FB15k-237 and WN18RR datasets. As shown, most entities have low degrees while only a small fraction exhibit high degrees. Based on these statistics, we select the bottom 20% of entities by degree as representatives of sparse entities and apply targeted data augmentation to these entities. According to our statistics, the bottom 20% of entities correspond to **2901** entities in FB15K-237 and **8,881** entities in WN18RR.

Because both datasets contain a large number of entities and large language models are susceptible to hallucinations, we designed a strategy to minimize noisy generations. Concretely, we adopt high thresholds for candidate selection to reduce the risk of presenting noisy inputs to the LLM. We also analyze the KG's topological patterns and restrict candidate entities to a small, high-confidence contextual subset. Details are reported in Table 7 and Table 8. Finally, we apply additional filtering

and validation to the generated triples: beyond removing triples that already appear in the training set, we enforce relation-specific heuristic rules to discard implausible triples (for example, for many-to-one relations we ensure generated tail entities are not duplicated). These measures improve the quality and reliability of the augmented data.

We also present an illustrative example of the candidate set (see Figure 8).

Table 6: Statistics of Long-Tail Distribution on Datasets. gini indicates the gini coefficient of entity degree distribution, and top80% indicates the percentage of entities whose degree is in the top 80% of all entities.

| Dataset | gini | top80% |
|---|---|---|
| FB15k-237 | 0.702 | 97.4% |
| WN18RR | 0.674 | 94.5% |

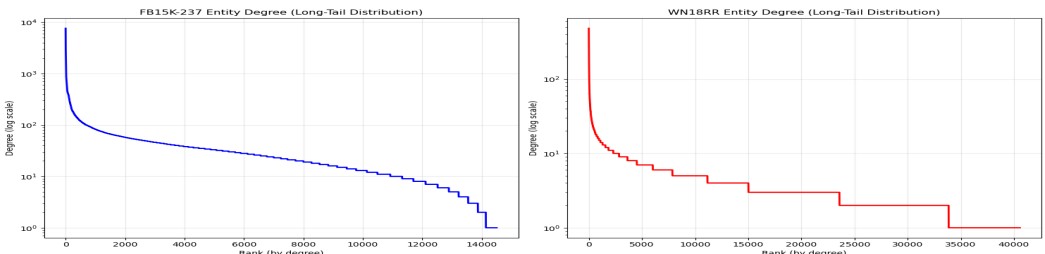

Figure 4: The long-tail distribution of entity degrees on FB15k-237 and WN18RR datasets. The x-axis represents entities sorted by their degree in descending order, while the y-axis indicates the degree of each entity.

Table 7: Distribution of shortest distance between head and tail entities in the training set of FB15k-237.

| Distance | Percentage | Cumulative% |
|---|---|---|
| 0 | 0.58 | 0.58% |
| 2 | 73.40 | 73.99% |
| 3 | 25.84 | 99.82% |
| 4 | 0.17 | 99.99% |

## B    DETAILS ABOUT GENERATED TRIPLETS

Table 9 presents statistics of the triples generated by the large language model. We further investigate the impact of the augmentation module on predictive performance for sparse entities. Table 10 reports a comparison of evaluation metrics for sparse entities in the FB15K-237 dataset before and after augmentation. The results indicate that augmentation yields substantial improvements across all reported metrics for sparse entities. Figure 5 illustrates the metric comparisons for sparse entities stratified by degree.

We present several example triples generated by the large language model. Figures 9 and 10 show generated triples that can be verified as correct (i.e., they appear in the test set). Figures 11 and 12 show uncertain triples that do not appear in the test set but are semantically plausible. Figure 13 illustrates erroneous (noisy) triples.

## C    USAGE OF LLMS AND PROMPT FOR KNOWLEDGE GRAPH TRIPLE GENERATION

In our experiments, we utilized the DeepSeek-V3 large-scale model for data augmentation. DeepSeek-v3 is a powerful multimodal large language model that demonstrates strong capabilities in textual understanding and reasoning. This prompt **get_possible_relations()** shown in Figure 6 is used in the relation generation stage to identify plausible relations when an entity serves as a head entity. Prompt **get_possible_tail_entity()** shown in Figure 7 is used in the entity generation stage to predict the correct tail entity given a head entity and a relation.

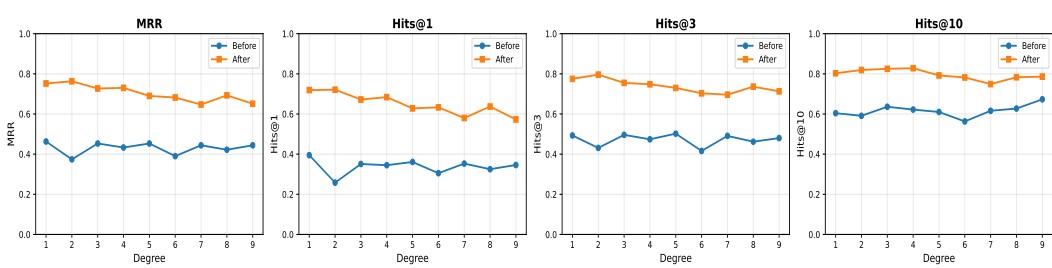

Figure 5: Metric comparisons for sparse entities stratified by degree in FB15K-237 before and after augmentation.

*The following is information related to a knowledge graph:*
*Entity: {entity_name}*
*Entity description: {entity_desc}*
*Known triplets related to entity {entity_name}:*
*({h}, {r}, {t})*
*...*
*Converting above known triplets into natural language: {nl}*
*Candidate relations:*
*{relation1},*
*{relation2},*
*...*
*Some known triplets corresponding to candidate relations:*
*({head}, {relation}, {tail})*
*...*
*Based on the above information, if the entity {entity_name} is used as the head entity, what are the possible relationships that could form valid triplets?*
*Please ensure that possible relationships are selected only from the candidate relations. Please return strictly JSON format where the key is possible relations and the value is number 1. If no possible relationships exist, strictly return an empty JSON format: {}.*

Figure 6: Prompt **get_possible_relations()** used in the relation generation stage.

*The following is information related to a knowledge graph:*
*Entity: {entity_name}*
*Entity description: {entity_desc}*
*Known context triplets related to entity {entity_name}:*
*({h}, {r}, {t})*
*...*
*Known triplets corresponding to relation {possible_relation}:*
*({h}, {r}, {t})*
*...*
*Candidate entities:*
*{candidate1}    entity_type:{type1}*
*{candidate2}    entity_type:{type2}*
*...*
*Consider the triple ({entity_name}, {possible_relation}, ?). Based on the information provided for {entity_name}, infer the appropriate tail entity or entities from the candidate list above. Ensure that each inferred triple is well justified by the available evidence.*
*If one or more valid tail entities can be inferred, return strictly a JSON object that maps each newly inferred triple to the integer 1, using the format:*
*{"(head,relation,tail)": 1}*
*If no valid tail entities can be inferred, return strictly an empty JSON object: {}*
*Do not include any additional text. Select tail entities only from the provided candidate set, and exclude any known triplets.*

Figure 7: Prompt **get_possible_tail_entity()** used in the entity generation stage.

```
{
    "/location/country/form_of_government": {
        "head_entity_types": [
            "Historical State",
            "Historical Event",
            "Location",
            "National Team",
            "Country",
            "Territory",
            "Place",
            "Historical Figure",
            "Historical Period",
            "Historical Entity",
            "Island",
            "Geopolitical Entity"
        ],
        "tail_entity_types": [
            "Government",
            "Concept",
            "Political System",
            "Government System"
        ],
        "head_entity_type_counts": {
            "Country": 212,
            "Location": 22,
            "Place": 30,
            "Historical Period": 3,
            "Geopolitical Entity": 3,
            "Historical Entity": 2,
            "Historical Figure": 2,
            "Territory": 4,
            "National Team": 1,
            "Historical State": 4,
            "Historical Event": 1,
            "Island": 1
        },
        "tail_entity_type_counts": {
            "Concept": 67,
            "Government": 114,
            "Government System": 51,
            "Political System": 9
        }
    }
}
```

Figure 8: Example of the candidate set for relation '/location/country/form_of_government' in FB15k-237.

Table 8: Distribution of shortest distance between head and tail entities in the training set of WN18RR.

| Distance | Percentage | Cumulative% |
|---|---|---|
| 1 | 48.02 | 48.02% |
| 2 | 8.77 | 56.80% |
| 3 | 19.37 | 76.16% |
| 4 | 6.67 | 82.83% |
| 5 | 8.10 | 90.94% |
| 6 | 3.83 | 94.77% |
| 7 | 2.62 | 97.38% |
| 8 | 1.85 | 99.23% |

Table 9: Statistics of triples generated by the LLM. "count" denotes the total number of generated triples; "count*" denotes the number of triples retained after filtering; "test-in" denotes the number of generated triples that also appear in the test set.

| Dataset | count | count* | test$_{in}$ |
|---|---|---|---|
| FB15k-237 | 10833 | 8956 | 932 |
| WN18RR | 12526 | 11789 | 198 |

## D COMPUTATIONAL COST OF THE MODEL

Table 11 presents a comparison of the computational costs associated with baseline models. It is worth noting that we restrict our comparisons to PLM-based models, as they generally achieve superior task performance but incur substantial computational overhead.

Table 12 reports the time overhead of performing data augmentation with large language models. FB15k-237 requires more time than WN18RR due to its larger number of relations and higher graph density.

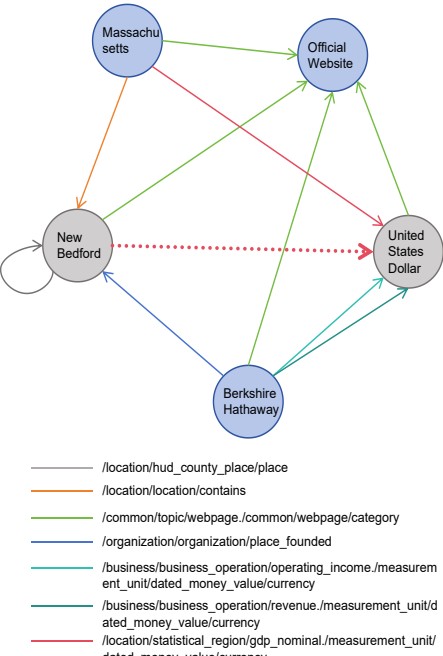

Figure 9: Example 1 of generated triples that can be verified as correct. Red dashed lines indicate triples generated by the large language model.

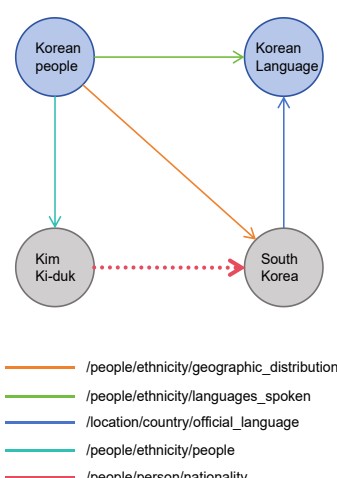

Figure 10: Example 2 of generated triples that can be verified as correct. Red dashed lines indicate triples generated by the large language model.

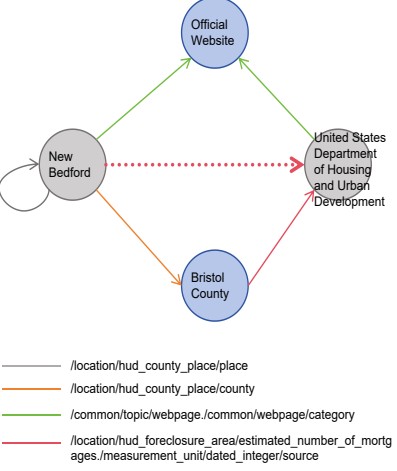

Figure 11: Example 1 of generated triples that are uncertain but semantically plausible. Red dashed lines indicate triples generated by the large language model.

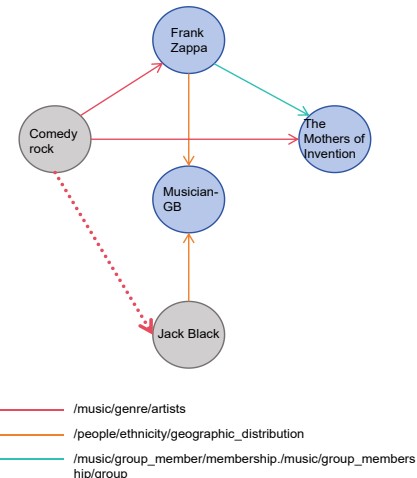

Figure 12: Example 2 of generated triples that are uncertain but semantically plausible. Red dashed lines indicate triples generated by the large language model.

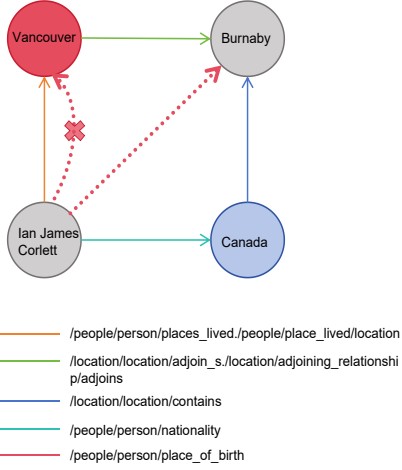

Figure 13: Example of generated triples that are erroneous (noisy). Red dashed lines denote ground-truth triples, while red crosses indicate erroneous triples produced by the large language model.

Table 10: Comparison of metrics for sparse entities in FB15K-237 before and after augmentation. "Before" denotes metrics prior to augmentation; "After" denotes metrics following augmentation.

|  | MRR | H@1 | H@3 | H@10 |
|---|---|---|---|---|
| Before | 0.427 | 0.337 | 0.465 | 0.606 |
| After | 0.695 | 0.624 | 0.735 | 0.810 |

Table 11: Computational cost comparison of different models. GPU indicates the type and number of GPUs used during training. Inference Time is measured on FB15K-237 test set.

| Model | GPU | No. of parameters | Training Time | Epochs | Inference Time |
|---|---|---|---|---|---|
| iHT | V100 / 16 | 110M | >60h | 5 | - |
| KG-BERT | - | 110M | - | - | - |
| RAA-KGC | - | >200M | - | - | - |
| KG-R3 | A6000 / 2 | 17.3M | 67h | 300 | - |
| **Ours** | 3060 / 1 | 19.3M (-gauss 15.1M) | 37h | 293 | 10.8s |

# E  EXPERIMENTAL RESULTS ON ADDITIONAL DATASETS

To validate the generality of our method, we conducted experiments on two additional datasets: UMLS and FB15K. UMLS is a small-scale biomedical knowledge graph containing 135 entities and 46 relations, while FB15K is a widely-used benchmark dataset, comprising 14,951 entities and 1,345 relations. The experimental results are presented in Table 13 and Table 14.

Figure 14 illustrates the long-tail distribution of entity degrees in the FB15k and UMLS datasets. Based on corresponding statistics, we selected less than 10% of entities for data augmentation experiments. The large language model generated 59 triples for UMLS and 3,026 triples for FB15K. Table 13 and Table 14 show that our model outperforms baseline methods on both datasets, demonstrating the effectiveness and generality of our approach.

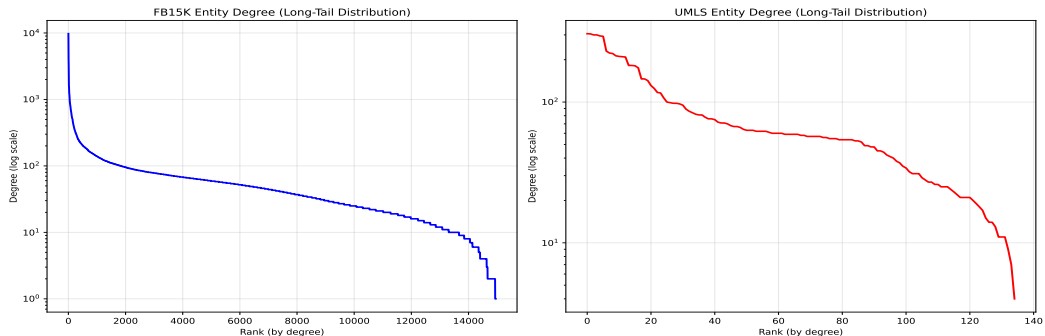

Figure 14: The long-tail distribution of entity degrees on FB15k and UMLS datasets.

Table 12: Timing statistics for data augmentation using large language models.

| Dataset | time | time per entity |
|---------|------|-----------------|
| FB15K-237 | 14h | 16s |
| WN18RR | 13h | 7s |

Table 13: Experimental results on UMLS dataset.

| Model | MRR | H@1 | H@3 | H@10 |
|-------|-----|-----|-----|------|
| RotatE | 0.744 | 0.636 | 0.822 | 0.939 |
| CompilE | 0.868 | 0.802 | 0.924 | 0.973 |
| KG-BERT | - | - | - | 0.990 |
| **Ours** | **0.899** | **0.826** | **0.969** | **0.990** |

Table 14: Experimental results on FB15K dataset.

| Model | MRR | H@1 | H@3 | H@10 |
|-------|-----|-----|-----|------|
| TransE | 0.463 | 0.297 | 0.578 | 0.749 |
| RotatE | 0.699 | 0.585 | 0.788 | 0.872 |
| QuatRE | 0.786 | 0.725 | 0.830 | 0.881 |
| ConvE | 0.657 | 0.558 | 0.723 | 0.831 |
| R-GCN | 0.696 | 0.601 | 0.760 | 0.842 |
| **Ours** | **0.803** | **0.732** | **0.865** | **0.917** |

