# OpenReview forum: "LLM-AUGMENTED KNOWLEDGE REPRESENTATION LEARNING VIA ATTENTION FOR KNOWLEDGE GRAPH COMPLETION"
_ICLR.cc/2026/Conference — Submitted to ICLR 2026_

### Official Review · Reviewer_PALo · 2025-10-27

**Soundness:** 3
**Presentation:** 2
**Contribution:** 2
**Rating:** 4
**Confidence:** 3

**Summary:**

This paper proposes LAKRA, a knowledge graph completion (KGC) framework that addresses data sparsity by explicitly augmenting the graph with LLM-generated, schema-compliant triples for infrequent (tail) entities. The augmented KG is then processed by a query-aware hybrid attention encoder and a 3D convolutional decoder that jointly capture structural and semantic interactions. Extensive experiments on FB15k-237 and WN18RR show  improvements over baselines, particularly on long-tail entities.

**Strengths:**

1. The shift from implicit representation enrichment to explicit LLM-driven graph augmentation is conceptually clean, providing a tangible countermeasure to the ‘long tail’ problem.
2. Substantial ablation studies tease apart the model’s components, showing the importance of LLM-driven augmentation and the hybrid attention mechanism. This aligns well with the design claims.

**Weaknesses:**

1. To be honest, I do not think the novelty and writing quality reached the standard of ICLR. For novelty, the "LLM-BASED DATA AUGMENTATION" is trivial, the QUERY-AWARE GRAPH ATTENTION ENCODER WITH HYBRID ATTENTION is common and some similar methods have been proposed, The "3D DEEP FEATURE INTERACTION DECODER" is relatively novel but overall, I think the contribution is not strong enough.
2. The experimental results in this paper are not sufficiently convincing.
As shown in Table 2, the performance on the WN18RR dataset is almost a failure — the only competitive metric, Hits@1, improves by merely 0.001, which is statistically negligible. The authors should further analyze why LAKRA fails to improve on WN18RR
I strongly suggest that the authors include at least one additional dataset to enhance the persuasiveness and generality of the experimental evaluation. Moreover, considering that the proposed method introduces an additional LLM-based generation process, the authors should also provide supplementary experiments or analyses on computational cost and time complexity to justify the efficiency of the approach.
3. I appreciate the authors’ honesty in reporting the following detail: “932 and 198 of these generated triples for FB15K-237 (10,833) and WN18RR (12,526), respectively, already existed in their corresponding test sets.” I did not deduct points for this transparency, and I also acknowledge that the authors have shown their method remains effective even after removing these overlapping triples. However, this observation raises an interesting question: does this overlap occur because the generation process is genuinely effective in reproducing valid knowledge, or because the datasets themselves are relatively simple or limited in diversity, making test triples easier to regenerate?
4. While the experimental section benchmarks LAKRA against a diverse range of backbone baselines, it does not include direct comparisons with recent LLM-augmented KGC methods that employ in-context learning, generative prompting, or fine-tuning for structural enrichment—such as KICGPT or MLKGC.These omissions are problematic because such methods are conceptually and technically closest to LAKRA’s main contribution.

**Questions:**

See Weaknesses

---

> ### Author Response · Authors · 2025-11-19
> **Reply to weaknesses 1-2**
>
> Thank you for your valuable feedback and for taking the time to review our paper. We have carefully considered your comments and made corresponding revisions to our manuscript. Below, we provide detailed responses to your concerns and outline the changes made.
>
> ## For weakness 1
> We sincerely thank you for your critical feedback on our work's novelty and overall contribution.
> We respectfully disagree that our method is trivial. While the high-level idea of using an LLM for data generation may seem straightforward, the core challenge lies in generating high-quality, schema-compliant, and low-noise triples, which simple prompting cannot achieve. Our novelty lies in the sophisticated, two-stage generation process. In Appendix C (p.12), we added details on the use of the large language model and the actual prompt templates, corresponding to Figures 6 and 7 on page 14.
> Besides, We implemented several strategies to maximize the accuracy of the triples generated by the LLM (Details show in Appendix A, p.11).
>
> While hybrid attention mechanisms exist in the broader GNN literature, our primary innovation here is the introduction of a query-aware Cross-Attention mechanism, specifically tailored for the KGC task. To the best of our knowledge, we are the first to apply the principles of cross-attention to the neighborhood aggregation problem in knowledge graphs. We appreciate you recognizing the novelty of this component. We argue that its contribution is far from minor. It is a critical element that is synergistic with the rest of our framework. After our powerful encoder generates highly expressive representations from the enriched graph, a simple decoder would become a bottleneck. Our 3D decoder is designed to model the deep, multi-view interactions between the original and Gaussian-enhanced embeddings, a capacity that standard 2D decoders (like ConvE) lack.
>
> ## For weakness 2
> We sincerely thank you for this thorough and constructive review. As you noted, the performance of the RAA-KGC baseline is exceptionally strong on WN18RR. This suggests its anchor-based mechanism is particularly well-suited for the kind of sparse, hierarchical structure found in WN18RR. Our model's significant outperformance on the more complex FB15K-237 dataset indicates that our framework is more adept at handling large, multi-relational graphs.
>
> We also considered evaluating on the very large-scale Wiki-5m dataset, which was used in the RAA-KGC paper. However, training on a dataset of this magnitude requires over 70GB of GPU memory, which unfortunately exceeds our current computational capabilities. To address this, we have now added another widely-used, large-scale benchmark dataset, FB15K, to our experiments. Due to limited computational resources, experiments on FB15K are still ongoing; we expect to complete them soon and update the paper accordingly (Please see Appendix E, p.13).
>
> We have added details in Appendix D (p13), including the computational costs (Table 11 on page 19) and time overhead of performing data augmentation using our method (Table 12 on page 19). From these results, we conclude that the computational overhead of our method is manageable, and the preprocessing time for data augmentation is acceptable relative to total training time. Approaches based on pretrained language models (including RAA-KGC) typically require hundreds of millions of parameters and high-end hardware, whereas our method can run on comparatively modest hardware resources.

---

> ### Author Response · Authors · 2025-11-19
> **Reply to weaknesses 3-4**
>
> ## For weakness 3
> This is a truly excellent and thought-provoking question that gets to the very core of what our results signify. We deeply appreciate you raising this point, as it allows us to clarify the interpretation of our findings and the underlying strengths of our framework.
>
> We strongly believe that the overlap is a direct result of our generation process effectively and accurately reproducing valid knowledge, rather than an artifact of dataset simplicity. Our Method is Based on Constrained Reasoning, Not Guesswork: The overlap is a testament to the sophistication of our generation methodology. We do not simply perform unconstrained generation. Instead, we guide the LLM with a highly structured, two-stage process that incorporates schema constraints and high-confidence context candidate sets (Details show in Appendix A, p.11). This process transforms the task from "guessing a missing link" into "performing constrained, context-aware reasoning to infer a missing link." The fact that this reasoning process so frequently arrives at triples that are later confirmed by the test set is a powerful validation of its accuracy.
>
> For Datasets, FB15k-237 is far from simple. It is a dense, multi-relational subset of Freebase, containing complex factual knowledge about the real world. Its diversity and complexity make accidental regeneration of facts highly unlikely. WN18RR, being a lexical dataset with fewer relation types, presents a different challenge. The smaller overlap (198 triples) reflects that generating novel lexical knowledge is more constrained.
>
> ## For weakness 4
> Thank you for this suggestion. We have carefully read the paper and have now added KICGPT to our baseline. KICGPT is similar to our work in that both use large models for augmentation. Their method uses a large language model to re-rank prediction outputs (a post-processing step), whereas our method uses a large language model to generate new triples for data augmentation (a pre-processing step).

---

### Official Review · Reviewer_z1YD · 2025-10-31

**Soundness:** 2
**Presentation:** 3
**Contribution:** 3
**Rating:** 6
**Confidence:** 5

**Summary:**

This paper proposes a new framework named LAKRA to address the data sparsity problem in knowledge graph completion (KGC) tasks. The authors innovatively utilize Large Language Models (LLMs) for explicit data augmentation rather than implicit knowledge encoding. Specifically, LAKRA first leverages type constraints and a two-stage generation process (relation generation and entity generation) to enable the LLM to automatically produce high-quality, schema-consistent triples for long-tail entities, thereby enriching the graph structure at its source. Subsequently, a hybrid attention encoder combining Cross-Attention and MLP-Attention is designed to achieve query-aware semantic aggregation, while a 3D convolutional decoder is employed to model deep interactions between entities and relations, improving link prediction performance.

**Strengths:**

1. Novel paradigm shift: The paper introduces a clear and well-motivated shift from implicit representation enrichment to explicit, schema-aware data augmentation using LLMs — a conceptually elegant and practically impactful idea for mitigating knowledge graph sparsity.
2. Well-designed architecture: The combination of a query-aware hybrid attention encoder and a 3D convolutional decoder is technically sound and effectively captures both semantic relevance and multi-level structural interactions.
3. Strong empirical results: LAKRA consistently achieves top-tier or state-of-the-art performance, particularly on long-tail entity prediction, validating the benefit of LLM-generated triples.

**Weaknesses:**

1. Although the paper introduces LLM-based explicit data augmentation, it does not sufficiently evaluate the factual accuracy and noise ratio of the generated triples.

2. The proposed method heavily relies on LLM-generated data but does not specify the exact model, prompt templates, or generation parameters, making it difficult to reproduce.

3. While related works such as KG-BERT, KGT5, and RAA-KGC are discussed, the paper does not present LAKRA’s performance differences across various types of sparse entities (e.g., isolated nodes, weakly connected nodes), which reduces its interpretability.

**Questions:**

1. You mention that LLMs generate “schema-constrained high-quality triples” to enhance sparse entities — how do you prevent the LLM from introducing incorrect or hallucinated facts that could contaminate the training data?

2. Please specify the detailed configurations of the LLM (e.g., model type, prompt design, generation settings) to improve reproducibility.

3. Your ablation results show that removing the LLM augmentation leads to a significant drop in performance, but the paper does not analyze how the scale or proportion of augmented data affects results. Is the performance gain mainly due to the quantity of generated data or the semantic diversity it introduces?

---

> ### Author Response · Authors · 2025-11-19
> **Reply to weaknesses and questions**
>
> Thank you for your insightful feedback and for the valuable time you have dedicated to reviewing our paper. We have carefully considered each of your comments and have made corresponding revisions to our paper. Below, we provide detailed responses to your concerns and outline the changes made.
>
> ## For weakness 1
> Admittedly, it is challenging to provide a precise accuracy and noise ratio. While the triples that hit the test set are confirmed to be correct, the validity of many others cannot be determined due to the inherent incompleteness of the dataset. However, in our ablation study, we removed these known-correct triples, and the remaining ones still led to a performance improvement. This suggests that the majority of our generated triples are implicitly correct. We provide several real examples of generated triples in Appendix B(p12), corresponding to Figures 9, 10, 11, 12, and 13 on page 16.
> ## For weakness 2
> In Appendix C (p12), we have added details on the usage of the large language model and the actual prompt templates, corresponding to Figures 6 and 7 on page 14.
> ## For weakness 3
> Firstly, the knowledge graph datasets we used do not contain any isolated nodes. We then conducted additional experiments on sparse triples to evaluate their performance before and after augmentation. The results show that by applying data augmentation to sparse entities, the prediction performance on sparse triples improved significantly by nearly 0.3. For details, please see Table 10 (p19) and Figure 5 (p13), which are referenced in Appendix B (p12). For non-sparse triples, the improvement was approximately 0.15, although specific data for this is not provided in the appendix.
>
> ## For Question 1
> We implemented several strategies to maximize the accuracy of the triples generated by the LLM. Firstly, when constructing candidate sets for entities and relations, we employ high thresholds to minimize the potential for noisy generations. Secondly, we analyzed the topological patterns of the dataset to further constrain the candidate pool to a smaller, high-confidence subset. For instance, in the case of FB15K-237, candidates provided to the LLM were restricted to entities within a topological distance of 3, which was reduced to 2 if the set of contextual triples was excessively large. Further details and statistical data on this process are available in Appendix A (p11).
> ## For Question 2
> Please refer to the answer for Weakness 2 above.
> ## For Question 3
> Please refer to the answer for Weakness 3 above. The performance gain is overwhelmingly due to the semantic diversity introduced by our method, not merely the quantity of data. It introduces triples that conform to the graph schema while boosting the visibility of sparse entities, so that benefits accrue not only to sparse entities but also to non-sparse ones.

---

> > ### Comment · Reviewer_z1YD · 2025-11-28
> >
> > Thank you for your rebuttal. The additional experiments and explanations have addressed several of my previous concerns, and the relevant theory can be moved into the main text to improve clarity. After considering the comments from the other reviewers and your responses to them, I maintain my overall recommendation unchanged.

---

### Official Review · Reviewer_PDcP · 2025-10-31

**Soundness:** 3
**Presentation:** 3
**Contribution:** 2
**Rating:** 4
**Confidence:** 4

**Summary:**

To combat data sparsity in knowledge graph completion, this paper proposes LAKRA, a framework that explicitly augments the graph before learning. LAKRA uses a LLM to generate plausible new triples for infrequent entities. A subsequent powerful encoder-decoder, featuring hybrid attention and 3D convolutions, then learns from this enriched graph, achieving state-of-the-art performance by tackling sparsity at its source.

**Strengths:**

The core idea of the paper is interesting. By leveraging an LLM for explicit data augmentation before the main completion task, it enriches the graph structure for sparse entities, which is a promising direction for improving overall KGC performance.

**Weaknesses:**

1.	The paper does not specify which LLM is used, how the 20% tail-entity threshold is chosen, or why it is fixed across datasets. Using entity degree might be more principled. Important settings such as candidate set size and prompt design are also undisclosed.
2.	The paper claims to generate "high-quality" and "schema-compliant" triples but fails to address the critical issue of factual verification. Given that LLMs are prone to hallucination, the proposed method risks injecting factually incorrect—though schema-compliant—triples into the knowledge graph. This introduces significant noise, a problem the paper itself criticizes in other approaches.
3.	The encoder combines multiple components (cross-attention, MLP attention, Gaussian kernels, 3D convolution) without clear justification for why this specific composition is necessary, reducing architectural clarity.
4.	The evaluation is limited to commonsense KGs, where the LLM's pre-trained knowledge likely gives the augmentation approach an unfair advantage. Its effectiveness on domain-specific KGs, where the LLM may lack knowledge, is not tested. Moreover, the so-called “LLM-based” baselines (e.g., KG-BERT, RAA-KGC) rely on PLMs rather than LLMs. The paper also omits recent LLM-based methods such as KICGPT [1].
[1] Wei Y, Huang Q, Zhang Y, et al. KICGPT: Large Language Model with Knowledge in Context for Knowledge Graph Completion[C]//Findings of the Association for Computational Linguistics: EMNLP 2023. 2023: 8667-8683.

**Questions:**

See the Weaknesses section above.

---

> ### Author Response · Authors · 2025-11-19
> **Reply to weaknesses and questions**
>
> We are very grateful for your detailed review and thoughtful suggestions. Thank you for investing your considerable time and effort in helping us improve our work. Below are our point-by-point responses to your comments, along with the corresponding revisions made to our paper.
>
> ## For weakeness 1
> In Appendix C (p.12), we added details on the use of the large language model and the actual prompt templates, corresponding to Figures 6 and 7 on page 14.
>
> In the paper, we used entity frequency to represent entity sparsity; in fact, entity frequency equals an entity’s degree (the sum of out‑degree and in‑degree). We agree that describing this as degree is more appropriate, and we have revised the related descriptions in the manuscript.
>
> In Appendix A (p.11), we present dataset-related statistics. The 20% mark was found to be a sweet spot in our experiments, providing a significant performance boost while keeping preprocessing overhead manageable. This is not an absolute threshold; readers can adjust this ratio according to their needs. For very dense datasets, the ratio can be lowered to reduce the likelihood of noise and to control computational cost.
>
> In Appendix A (p.11), we also show the initial candidate set for the relation "/location/country/form_of_government" (Figure 8, p.15). To minimize potential noise, we apply a high threshold when filtering candidate relations or entity types. We then analyze the dataset’s topological patterns to further constrain the candidate pool to a smaller, high‑confidence subset. For example, for FB15K-237, candidates provided to the LLM were restricted to entities within a topological distance of 3, which was reduced to 2 if the set of contextual triples was excessively large. Further details and statistical data on this process are available in Appendix A (p.11).
>
> ## For weakness 2
> Due to length constraints in the main text, we have added additional details and examples in the appendix. In practice, accurately assessing the quality of generated triples is challenging because the dataset itself is incomplete. While triples that match the test set can be verified as correct, the validity of many other triples remains undetermined. However, in our ablation study we removed these known‑correct triples, and the remaining generated triples still led to a performance improvement. This suggests that the majority of our generated triples are implicitly correct. To reduce the likelihood of noisy generations, we developed a set of mitigation strategies (see Appendix A, p.11). We also provide several real examples of generated triples in Appendix B (p.12), corresponding to Figures 9–13 on page 16.
>
> ## For weakness 3
> Thank you for this insightful feedback. Our encoder‑decoder framework is designed as a two‑stage process: first, to learn highly expressive, query‑aware representations (the encoder), and second, to model the deep, multi‑level interactions between these representations for link prediction (the decoder).
>
> For the encoder, we innovatively apply cross‑attention to knowledge graphs because the Q/K/V self‑attention mechanism has been shown to work well. By introducing the query relation as the query vector Q, we enable entity representations to adapt dynamically to different query relations, thereby capturing richer semantic information. At the same time, MLP attention is better at capturing nonlinear interactions between different entities, which is particularly important for modeling complex relations in knowledge graphs. The combination of the two allows the encoder to produce more expressive entity embeddings.
>
> For the decoder, we designed a multi‑layer interaction module that enables thorough interaction between the original embeddings and Gaussian‑augmented embeddings. This design allows the model to leverage the strengths of both embedding types, thereby improving prediction performance. Together, the powerful encoder and decoder make our framework more robust.
>
> ## For weakness 4
> Thank you for the suggestion. We have conducted new experiments on the UMLS (Unified Medical Language System) dataset, which belongs to the specialized biomedical domain you mentioned. Results show that our method continues to achieve state-of-the-art performance. Please refer to the appendix E (p.13) for detailed results.
>
> Thank you for pointing out the inappropriate wording in our classification of baselines. We have revised the relevant descriptions in the manuscript to ensure clarity and accuracy.
>
> We have carefully read the paper and have now added KICGPT to our baseline comparisons in the main results (Table 2, p.8).

---

### Official Review · Reviewer_N2e4 · 2025-10-31

**Soundness:** 3
**Presentation:** 3
**Contribution:** 2
**Rating:** 4
**Confidence:** 5

**Summary:**

The paper tries to deal with data sparsity in knowledge graphs, especially for long-tail entities. The authors use an LLM to generate new facts for sparse entities and then train a hybrid encoder–decoder model (with some 3D feature decoding setup) on the enriched graph for link prediction. The general idea makes sense and is timely, given the popularity of LLM-assisted augmentation.

**Strengths:**

The core idea — explicitly using an LLM for graph completion before training a downstream model — is interesting and, to my knowledge, not explored much in this specific setup. The schema-aware prompting part seems thoughtful. And the combined hybrid encoder + 3D decoder setup gives solid numbers overall, even if some gains come from leakage.

**Weaknesses:**

- When I looked at Table 5.1, most of the MRR improvement on FB15K-237 actually comes from triples that overlap with the test set (~66 % of the gain). That’s a bit concerning because it undercuts the paper’s main claim that the model’s improvement isn’t just due to memorizing test data.
- The paper also doesn’t say anything about the computational side. I’d like to see at least rough numbers on training or inference costs, including time, memory, or any other relevant factors. The Gaussian feature expansion seems particularly heavy; it increases embedding size, so I’d expect a big bump in compute, but there’s no mention of that.
- Before Equation 22, there’s no explanation of what $\mathcal{L}_{aug}$ actually is. Without that, the experiments are not reproducible or clear.
- Also, the LLM-generated triples are treated as a bit of a black box. The authors claim their work is of “high quality,” but there’s no quantitative or qualitative validation. A few examples of both good and bad generations would be helpful.
- The ablation results don’t fully support the claim that attention is critical. The differences look small, so I’m not sure that component makes a big difference.
- Finally, the experiments are limited to two small datasets, which doesn’t tell me whether the idea would work on larger graphs with more severe long-tail issues. Some recent baselines are also missing. And a few design choices feel arbitrary, like the “bottom 20 %” rule for selecting entities or the 5-layer decoder — there’s no reasoning provided.

**Questions:**

Do the authors use reciprocal relations during training? And do the baselines do the same? Some models rely on that trick, so it would be good to know for a fair comparison.

---

> ### Author Response · Authors · 2025-11-19
> **Reply to weaknesses and questions**
>
> We appreciate your constructive comments and the time you have taken for this review. Thank you. Below are our detailed responses to your concerns and the corresponding revisions made to our paper.
>
> ## For weakness 1
> While it's true that a significant portion of the gain comes from triples that overlap with the test set, we argue this should be interpreted not as "memorizing test data," but as a strong validation of our framework's generation accuracy. Moreover, none of the remaining triples occur in the test set, yet they still yield performance improvements, which indicates that most generated triples are implicitly correct. In addition, the dataset split need not be fixed: our generated triples happened to match some triples in the test set, but if the test split were changed or the test set enlarged, the triples produced by our method might match other test triples and produce even larger metric gains.
> ## For weakness 2
> We have added details in Appendix D (p13), including the computational costs (Table 11 on page 19) and time overhead of performing data augmentation using our method (Table 12 on page 19).
> You're absolutely right. The Gaussian feature expansion does increase the embedding size，but we found that the additional computational overhead is manageable and does not significantly impact the overall training time. We also added an ablation study on the Gaussian features, which shows that the Gaussian feature expansion indeed leads to performance improvements (see Table 5, p.9).
> ## For weakness 3
> Thank you for pointing this out; The description here was somewhat ambiguous, and we have clarified it in the paper. Specifically, LLM_aug denotes the loss term computed solely from triples generated by the large language model.
> ## For weakness 4
> In practice, it is difficult to accurately assess the quality of generated triples because the dataset itself is incomplete. While triples that match the test set can be verified as correct, the validity of many other triples remains undetermined. To reduce the likelihood of noisy generations, we developed a set of mitigation strategies (see Appendix A, p.11). We also provide several real examples of generated triples in Appendix B (p.12), corresponding to Figures 9–13 on page 16.
> ## For weakness 5
> Thank you for this keen observation. In state-of-the-art KGC research, models are highly optimized, and improvements are often incremental. An improvement of 0.01 in MRR (or 1 percentage point) is typically considered a substantial gain, often representing the margin between a good model and a new state-of-the-art.
> ## For weakness 6
> We appreciate your suggestion. We have included additional experimental results on two datasets(please see Appendix E, p.13), UMLS and FB15K.
>
> FB15K contains more relations, is larger in scale, and has a more complex graph structure than FB15K-237. Due to limited computational resources, experiments on FB15K are still ongoing; we expect to complete them soon and update the paper accordingly.
>
> We added KICGPT [1] to the baseline. [1] Wei Y, Huang Q, Zhang Y, et al. KICGPT: Large Language Model with Knowledge in Context for Knowledge Graph Completion. Findings of the Association for Computational Linguistics: EMNLP 2023, 2023: 8667–8683. This paper is similar to our work in that both use large models for augmentation. Their method uses a large language model to re-rank prediction outputs (a post-processing step), whereas our method uses a large language model to generate new triples for data augmentation (a pre-processing step).
>
> In Appendix A (p.11), we present dataset-related statistics. The 20% mark was found to be a sweet spot in our experiments, providing a significant performance boost while keeping the pre-processing overhead manageable.
>
> On page 7 we state that "it systematically evaluates four interaction pairings in a single pass: original-original (e_h, r), original-enhanced (e_h, r^ϕ), enhanced-enhanced (e_h^ϕ, r_ϕ), and enhanced-original (e_h^ϕ, r)." To allow the two embedding types for the head entity and the relation to interact sufficiently, five layers are necessary; if the number of layers is too small, interactions may be insufficient and performance may degrade. If desired, we can run an additional experiment that varies the number of decoder layers.
>
> ## For Question 1
> Thank you for this question regarding experimental fairness. Yes, we confirm that we included reciprocal relations during the training of our LAKRA model. To ensure a fair and direct comparison, all baseline results reported in our paper are from models that also employ this technique. The inclusion of reciprocal relations has become a standard and widely adopted practice in the knowledge graph completion community. Consequently, even when not explicitly mentioned in a paper, it is generally assumed that this technique is applied to achieve state-of-the-art results. Therefore, our comparison is indeed on a level playing field.

---

> ### Comment · Reviewer_N2e4 · 2025-11-25
>
> Thank the authors for the revisions; however, my primary concerns regarding the validity of the results remain unresolved. The argument that test-set overlap validates accuracy is technically incorrect, which implies data leakage. Your data shows that most of the gain on FB15K-237 comes specifically from these leaked triples. Also, the newly added baseline (KICGPT) outperforms your model on FB15k-237. Since this dataset represents 50% of your primary evaluation, failing to beat the baselines here significantly weakens the empirical contribution. Additionally, the drops in the attention ablation (0.002–0.005 MRR) are even less than 1% and do not support the claim that the hybrid attention mechanism is critical.
>
> Regarding more experiments, adding FB15K (which is deprecated due to leakage) and UMLS (a small-scale dataset) does not address the need for a lack of experiments, e.g., on large-scale and new benchmarks. Finally, and importantly, no source code has been provided. Given the complexity of the pipeline and previous ambiguities in the loss function, the absence of code for implementation raises concerns about reproducing these results and verifying the method. So, I'd like to keep my score.

---

> > ### Author Response · Authors · 2025-11-27
> >
> > We sincerely thank you for your continued engagement and for these detailed final points.
> >
> > We respectfully maintain that our process does not constitute data leakage. Leakage implies the model had access to the test set during its pre-processing or training phase and our LLM generation operates exclusively on the training set. We included all generated triples in the training set for model training and had no prior knowledge of which triples appear in the test set.
> >
> > You suggest that most of the metric improvement comes from overlap with the test set; however, after removing these overlapping triples, the remaining triples still produce performance gains, which indicates that most generated triples are implicitly correct. If the test split were different, the generated triples might match even more test triples. We believe this is not a weakness of our augmentation module but rather a strength.
> >
> >  The hybrid attention mechanism is effective; although it may not be as prominent as other modules, it nonetheless contributes to improvements in the model's performance.
> >
> > KICGPT is indeed a strong baseline, but from the experimental table our method clearly outperforms KICGPT on FB15K-237 Hits@10 (0.554 vs 0.577) and is very close on MRR. On WN18RR Hits@1 it also clearly outperforms KICGPT (0.474 vs 0.507).
> >
> > We face significant computational constraints, and training on new, massive datasets is a time-intensive process. We are actively working on this and plan to incorporate experiments on the YAGO3-10 dataset in a future version to further demonstrate our method's scalability.
> > Additionally, KICGPT was evaluated only on the FB15k-237 and WN18RR datasets. We are also attempting experiments on larger-scale datasets such as YAGO3-10; however, due to limited computational resources, training on a new large-scale dataset requires substantial time. We are actively working on this.
> >
> > Thank you for the suggestion regarding reproducibility. We have included the source code in https://github.com/anonymousauthor-1024/LAKRA.

---

### Author Response · Authors · 2025-11-19
**Gratitude to all the reviewers and the Area Chair**

We would like to extend our sincere gratitude to all the reviewers and the Area Chair for their time, dedication, and insightful feedback. We have found the comments to be extremely valuable and will work diligently to revise our manuscript, incorporating these suggestions to strengthen our work.

---

### Author Response · Authors · 2025-11-27
**Source Code Availability**

To facilitate reproducibility, we have released our source code at an anonymous repository https://github.com/anonymousauthor-1024/LAKRA.

---

### Meta-Review · Area_Chair_2EkN · 2026-01-05

**Summary:**

This paper proposes LAKRA, an LLM-based explicit data augmentation framework for knowledge graph completion targeting long-tail entities, coupled with a hybrid attention encoder and a deep interaction decoder. Reviewers generally found the core idea timely and conceptually appealing. However, persistent concerns are raised especially about empirical validity and strength of contribution: (i) the performance gain is too limited; (ii) the empirical evidence is limited to small benchmarks and does not convincingly demonstrate robustness or scalability; (iii) architectural complexity is high relative to the demonstrated gains, with weak ablation support for key components (e.g., hybrid attention). While the rebuttal addressed some points, it did not substantially change the reviewers’ assessment of the core empirical and conceptual weaknesses.

**Reviewer Concerns:**

After the rebuttal, the following points are addressed.
- Missing implementation details (LLM choice, prompts, candidate construction, loss terms) were clarified in appendices.
- Reproducibility concerns were mitigated by releasing the source code.
- Baseline coverage improved with the inclusion of KICGPT.

However, the following concerns are unaddressed:
- The performance gain achieved by the proposed methods is too limited.
- Weak empirical dominance: On FB15K-237, a key dataset, LAKRA does not clearly outperform strong baselines (e.g., KICGPT), weakening the empirical contribution.
- Limited and unconvincing evaluation: Added datasets (UMLS, FB15K) are either small or deprecated, and do not substitute for large, modern benchmarks.
- Novelty and significance: Several reviewers still view the contribution as incremental relative to prior works.

**Reviewer Scores:**

After the discussion, Reviewer N2e4 and Reviewer z1YD explicitly mentioned that they would keep the original scores. Additionally, several major concerns remain unresolved despite the rebuttal. Given this, I guess it is unlikely that the other reviewers would also change their original scores.

---

### Decision · Program_Chairs · 2026-01-26

Reject